# Dysfunctional Sars-CoV-2-M protein-specific cytotoxic T lymphocytes in patients recovering from severe COVID-19

Hideki Ogura [1] ✉, Jin Gohda[2], Xiuyuan Lu [3], Mizuki Yamamoto [2], Yoshio Takesue [4,5], Aoi Son [1], Sadayuki Doi[6], Kazuyuki Matsushita[7], Fumitaka Isobe[8], Yoshihiro Fukuda[9], Tai-Ping Huang[10], Takamasa Ueno [11], Naomi Mambo[12], Hiromoto Murakami[12], Yasushi Kawaguchi [2,13], Jun-ichiro Inoue [14], Kunihiro Shirai[12], Sho Yamasaki [3,15,16,17], Jun-Ichi Hirata[12] & Satoshi Ishido [1] ✉

Although the importance of virus-specific cytotoxic T lymphocytes (CTL) in virus clearance is evident in COVID-19, the characteristics of virus-specific CTLs related to disease severity have not been fully explored. Here we show that the phenotype of virus-specific CTLs against immunoprevalent epitopes in COVID-19 convalescents might differ according to the course of the disease. We establish a cellular screening method that uses artificial antigen presenting cells, expressing HLA-A*24:02, the costimulatory molecule 4-1BBL, SARS-CoV-2 structural proteins S, M, and N and non-structural proteins ORF3a and nsp6/ORF1a. The screen implicates SARS-CoV-2 M protein as a frequent target of IFNγ secreting CD8$^+$ T cells, and identifies M$_{198–206}$ as an immunoprevalent epitope in our cohort of HLA-A*24:02 positive convalescent COVID-19 patients recovering from mild, moderate and severe disease. Further exploration of M$_{198–206}$-specific CD8$^+$ T cells with single cell RNA sequencing reveals public TCRs in virus-specific CD8$^+$ T cells, and shows an exhausted phenotype with less differentiated status in cells from the severe group compared to cells from the moderate group. In summary, this study describes a method to identify T cell epitopes, indicate that dysfunction of virus-specific CTLs might be an important determinant of clinical outcomes.

Even at present, SARS-CoV-2 is spreading across the world and causing COVID-19 with diverse clinical features; severe/critical respiratory infections such as pneumonia and acute respiratory distress syndrome is fatal outcomes especially in elder patients[1]. Therefore, the mechanisms behind severe outcomes have been intensively investigated[2,3]. Among these, an inability to engage timely anti-viral immune responses is proposed[4,5]. In general, type I interferon (IFN-I) suppresses viral propagation, supports tissue repair and stimulates adaptive immunity, so IFN-I plays a critical role in innate immunity[6]. This is the case with SARS-CoV-2 infection[7,8]; in some critically ill

COVID-19 patients, insufficient IFN-I activation was reported[9]. In this scenario, impaired IFN-I activation leads to incomplete elimination of virus followed by subsequent leukocyte-related inflammation, the so-called two-step model of pathogenesis[5].

As mechanisms other than impaired IFN-I signaling, maladapted acquired immunity is also proposed[10,11]. Among essential cells in acquired immunity against COVID-19, cytotoxic T cell (CTL) and B cells were demonstrated to provide sterilizing immunity and humoral immunity, respectively[12,13]. Autoreactive B cells have been reported to produce autoantibody against IFN-I, which suppress IFN-I signaling[14–16].

Profound depletion of T cells in peripheral bloods was present presumably due to overproduction of inflammatory cytokines[17]. Thus, dysregulation of both innate and acquired immunity could be associated with COVID-19 severity. However, it is not fully understood how differentiation status of antigen/virus-specific CTLs is related to disease severity. Since antigen/virus-specific CTLs are essential sentinels against invading pathogens, thorough characterization of these in COVID-19 patients is thought to be necessary to unveil the mechanisms of COVID-19 progression, as well as to inform therapeutic strategies.

In this work, we explore immunoprevalent epitopes from COVID-19 convalescents for examining SARS-CoV-2-specific CTLs. We perform deep and thorough analysis to characterize the SARS-CoV-2 immunoprevalent epitope-specific CTLs for searching the features related to disease severity. The identified traits of dysfunction of the cells of severe COVID-19 convalescents highlight the impaired virus-specific CTL development as a possible determinant of clinical outcomes.

## Results

### Generation of artificial antigen-presenting cells
To explore and better define SARS-CoV-2-specific CD8+ T cell responses in convalescents, we sought to prepare artificial antigen-presenting cells (aAPCs) stably expressing SARS-CoV-2 proteins. Human myeloid leukemia K562 cells were employed as parental cells as we previously observed minimum antigenicity of K562 to human CD45RO+CD8+ T cells[18]. Since 70.6% (36/51) of COVID-19 convalescents (mild, 60.0% (12/20); moderate, 84.6% (11/13); and severe, 72.2% (13/18)) and 75.0% (9/12) of healthy controls enrolled in this study carried an HLA-A*24:02 allele, K562 cells were firstly engineered to express HLA-A*24:02, and the costimulatory molecule 4-1BBL (Supplementary Fig. 1a, b)[18,19]. The function of antigen presentation was confirmed using influenza virus and cytomegalovirus (Flu/CMV)-derived peptides and peptide-dependent IFNγ secretion from a specific T cell line (Supplementary Fig. 1c). Then, three SARS-CoV-2 structural proteins (S, M, and N) and two non-structural proteins (ORF3a and nsp6/ORF1a), which were reported as viral antigens of SARS-CoV-2-specific CD8+ T cells, were selected to be expressed in aAPCs[20]. Stable expression of each SARS-CoV-2 protein was confirmed by quantitative RT-PCR and western blot (Fig. 1d, e). Because these aAPCs endogenously express viral proteins, the proteins are processed and presented as viral peptides associated with MHC class I via the intracellular antigen processing pathway[21]. With these aAPCs, a new screening system was established (Fig. 1a).

### M protein is an immunocompetent viral protein
In order to achieve an in-depth analysis of SARS-CoV-2-specific CD8+ T cells, we generated CD8+ T cell libraries from peripheral blood cells of COVID-19 convalescents. These libraries are useful for the definitive characterization of low frequency, but potentially crucial, CD8+ T cell populations in peripheral blood because rare, virus-specific CD8+ T cells are polyclonally expanded followed by stimulation with relevant antigen-expressing aAPCs (Fig. 1a). COVID-19 convalescents enrolled in this study were admitted to, or consulted with a physician, at hospitals affiliated with Hyogo College of Medicine or Kyowa-kai Medical Corporation from late 2020. Convalescents had experienced mild, moderate, or severe COVID-19 (Supplementary Table 1). Disease severity of COVID-19 was classified according to a recent report[22] (see Methods). Since HLA-A*24:02 was the dominant allele in our study group, we focused on HLA-A*24:02+ subjects and a total of 36 COVID-19 convalescents and 9 healthy volunteers were enrolled in the study (Supplementary Table 1), and 20 convalescents and 8 healthy volunteers among them were subjected to the library assay. A small number (two thousand) CD45RO+ CD8+ T cells isolated from each participant's PBMC were seeded into each well of 96-well-plates and polyclonally expanded with PHA in the presence of allogenic PBMC and cytokines to establish a library (Fig. 1a). All the established libraries from each participant were evenly divided into 7 groups (i.e., 5 groups for a series

of SARS-CoV-2 proteins, 1 group for mixed peptides derived from influenza virus and cytomegalovirus, and 1 group for a non-antigen negative control) and incubated with relevant aAPCs individually. IFNγ in the culture supernatant was measured as a functional read-out for antigen-specific CD8+ T cells, because it is a major effector cytokine of various virus-specific human CD8+ T cells[23–27] and because IFNγ+ CD8+ T cells are strongly associated with low disease severity among acute cases of COVID-19[28]. The response to the antigens was defined as positive when the IFNγ level was above mean +3 SD of that in the negative control (i.e., incubation with parental aAPCs). In total, 5403 libraries were examined in COVID-19 convalescents and healthy volunteers (3808 and 1595, respectively). Frequencies of IFNγ−positive libraries upon each antigen stimulation were compared among participant groups (i.e., COVID-19 convalescents and healthy volunteers). There were minimum responses observed in the non-antigen negative control groups of healthy and COVID-19 subjects; representative data are shown in Fig. 1b. Consistent with the previous reports, the response was observed against various antigens including structural and non-structural proteins (Supplementary Figs. 2, 3). Of note, those included reported immunodominant epitopes: $S_{1208-1216}$[29] and $ORF3a_{112-120}$[30] (Supplementary Fig. 3b). Importantly, SARS-CoV-2-M protein induced high responses in the libraries of convalescents compared with those of healthy volunteers (p = 0.0019, Mann–Whitney U test) (Fig. 1c, d) (Fig. 2) in our cohort. Interestingly, the frequency is the highest in the convalescents recovered from moderate severity COVID-19 (Fig. 1c, e). Although pre-existing cross-reactive immune memory to SARS-CoV-2 has been suggested[31], no significant responses observed against the proteins we tested were detectable in samples from healthy subjects of the cohort.

### Identification of an immunoprevalent CTL epitope $M_{198–206}$
To further examine the importance of M protein-specific CD8+ T cells in the context of SARS-CoV-2 infection, we sought to identify immunodominant epitopes of the M protein. The candidate M protein epitopes were obtained in silico as follows: candidate binders to HLA-A*24:02 were screened using full-length M protein amino-acid sequence (YP_009724393.1) and the Immune Epitope Database (https://www.iedb.org), and the top 5 peptides with the highest score (0.5 <Score, Percentile rank <0.2) and having a typical amino acid length were chosen (Table 1). Once M protein-responding libraries were obtained, they were further expanded and divided into multiple wells to examine the response to each candidate peptide by coculturing with aAPCs pulsed with the individual peptide. Next, the supernatant was collected to measure IFNγ levels again. For each candidate peptide, the response was defined as positive when IFNγ levels were statistically higher than those of controls (i.e., stimulation with aAPCs alone) and the mean of IFNγ levels was more than twice that of controls. The majority of the libraries responded to only a single peptide, but some responded to two or more peptides, indicating the presence of two or more SARS-CoV-2-M-specific T cell clones among the 2000 original CD45RO+CD8+ T cells. In total, 37 libraries responded to M protein-expressing aAPCs (aAPC-M) from the 7 convalescents were tested for their antigen specificity. Surprisingly, 81.1% (30/37) of the libraries were found to respond to the $M_{198–206}$ peptide (Fig. 2a, b). Moreover, the response to $M_{198–206}$ was detected in the libraries from 6 out of 7 convalescents (Fig. 2b, c). In addition, in 5 out of 6 convalescents whose libraries responded to $M_{198–206}$, the frequency in the $M_{198–206}$ responding library was more than half (Fig. 2c). Lastly, the presence of $M_{198–206}$-specific CD8+ T cells in the libraries was confirmed by staining with MHC class I tetramer generated with HLA-A*24:02 and synthetic $M_{198–206}$ peptide (Fig. 2d). By confirming its direct association to HLA-A*24:02 by competition assay (Supplementary Fig. 4), we concluded that $M_{198–206}$ is one of the SARS-CoV-2 major epitopes in M protein. In order to examine the reliability of the library assay, we performed

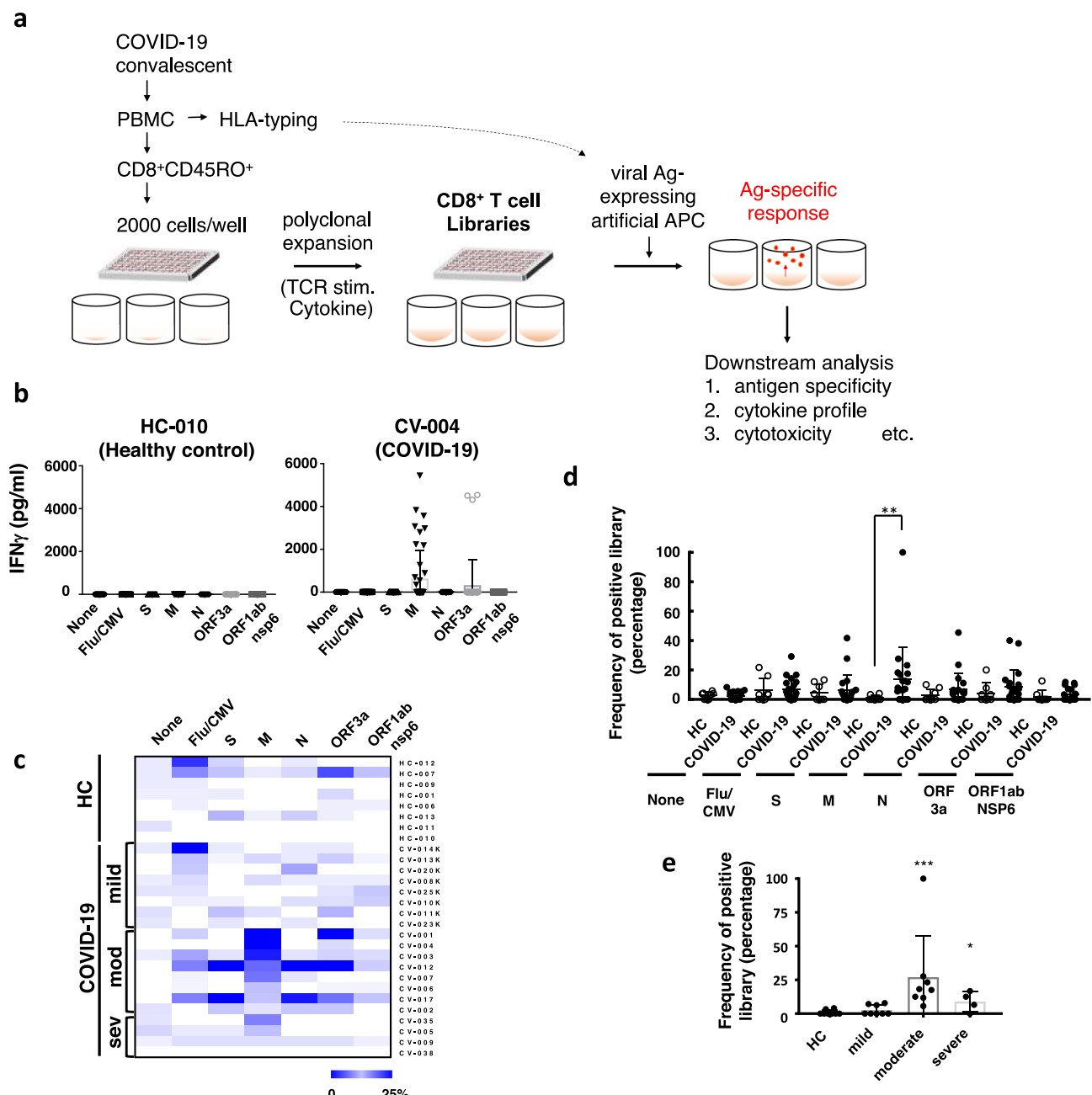

**Fig. 1 | Cellular screening of SARS-CoV-2-specific response. a** Memory CD8+ T cells (CD45RO+) from peripheral blood mononuclear cells (PBMC) of COVID-19 convalescents were seeded at 2000 cells/well in multiple wells containing irradiated allogenic PBMC, PHA, IL-2, IL-7, and IL-15 as described in Methods. The individual cultures were expanded to make up the libraries of polyclonally amplified CD8+ T cells. Then the libraries were washed and examined as to their antigen specificity by stimulating with a series of viral protein-expressing aAPC. Once being judged as positive by IFNγ level (measured by ELISA), they were further expanded with relevant aAPC and cytokines. These libraries (cell lines) were subjected to several downstream analyses including epitope screening, cytokine profiling, and/ or cytotoxicity assays, etc. **b** A representative result of the CD8+ T cell library assay. Libraries from a healthy donor (HC-010) or COVID-19 convalescent (CV-004) were divided into 7 groups, and each group was co-cultured with indicated aAPCs: S ($n = 15$ for HC-010; $n = 47$ for CV-004), M ($n = 15$ for HC-010; $n = 47$ for CV-004), N ($n = 15$ for HC-010; $n = 47$ for CV-004), ORF3a ($n = 14$ for HC-010; $n = 41$ for CV-004), or ORF1ab NSP6 ($n = 14$ for HC-010; $n = 53$ for CV-004) -expressing aAPCs, influenza virus and cytomegalovirus (Flu/CMV) derived peptides ($n = 15$ for HC-010; $n = 47$ for

CV-004)-pulsed aAPC or aAPCs (None) ($n = 15$ for HC-010; $n = 47$ for CV-004). Each dot represents the IFNγ level of each library. The threshold of positivity was determined by mean + 3 SD of IFNγ level in the group co-cultures with aAPCs (indicated as "None"). **c** Heatmap representation of results from all participants (healthy controls (HC), convalescents from mild, moderate (mod), and severe (sev) disease) subjected to the library assay. The frequency of positive library in response to aAPCs (None), indicated viral protein-expressing, or influenza virus and cytomegalovirus (Flu/CMV) derived peptides-pulsed aAPC are shown. **d** Comparison of positive library frequency between healthy donors (HC) (open circles, $n = 8$) and COVID-19 convalescents (COVID-19) (filled circles, $n = 20$) in each group. $p$ values were calculated by two-sided Mann–Whitney test. Data represent mean ± SD. **$p = 0.0064$. **e** Comparison of positive library frequency in response to SARS-CoV-2-M protein-expressing aAPC among indicated groups (healthy donors (HC) ($n = 8$), convalescents from mild severity COVID-19 (mild COVID-19) ($n = 8$), those from moderate severity COVID-19 (moderate COVID-19) ($n = 8$) and those from severe severity COVID-19 (severe COVID-19) ($n = 4$). $p$ values were calculated by two-sided Mann–Whitney test. Data represent mean ± SD. ***$p = 0.0002$; *$p = 0.024$.

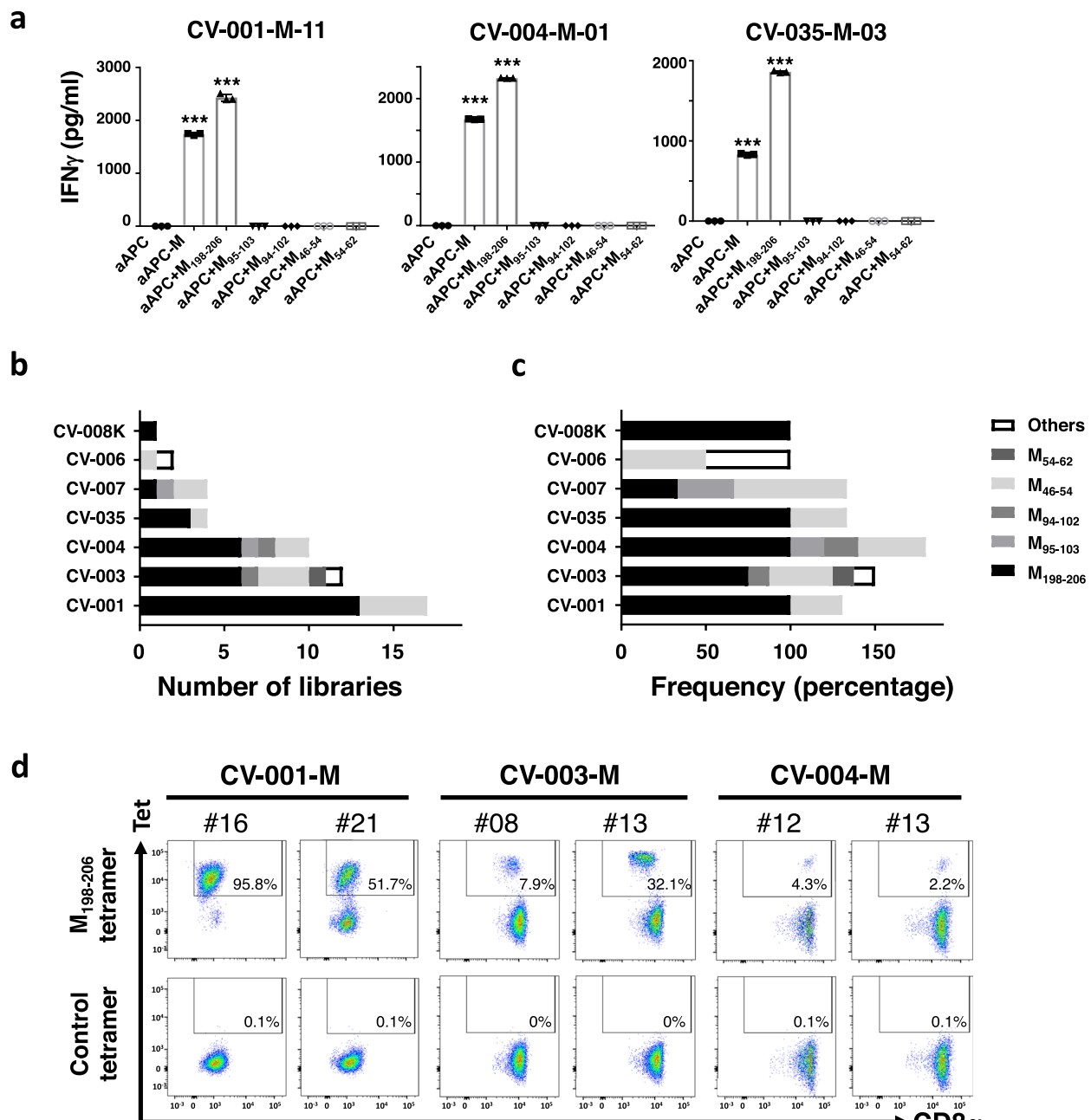

**Fig. 2 | Identification of M198−206 as a major epitope. a** Representative data from epitope screening. The libraries showing positive responses to SARS-CoV-2-M protein-expressing aAPC (aAPC-M) were further expanded, divided into seven groups, and co-cultured with indicated aAPCs: aAPC, aAPC-M or each peptide-pulsed aAPCs. $M_{198–206}$, $M_{95–103}$, $M_{94–102}$, $M_{46–54}$ and $M_{54–62}$ peptides were tested as epitope candidates. IFNγ levels in supernatant of libraries co-cultured with indicated aAPCs are shown. Data represent mean + SD ($n = 3$ in each group). $p$ values were calculated with two-sided unpaired $t$ test. ***$p = 1.2 \times 10^{-8}$, $3.4 \times 10^{-7}$ for CV-001-M-11; ***$p = 9.8 \times 10^{-11}$, $5.0 \times 10^{-12}$ for CV-004-M-01; ***$p = 1.2 \times 10^{-8}$, $1.2 \times 10^{-9}$ for CV-035-M-03. **b, c** The number (**b**) and frequency (**c**) of library which was positive in response to each peptide are shown in each COVID-19 convalescent.
**d** Representative data of flowcytometric analysis for M protein-responding libraries from COVID-19 convalescents with $M_{198–206}$-MHC tetramer or control tetramer.

activation induced marker (AIM) assay with the same set of M peptides. As expected, $M_{198–206}$ response was observed in the convalescents having M-responding libraries (CV-004, and CV-007, Supplementary Fig. 5), but not in the one having no responding library (CV-006, Supplementary Fig. 5). There was minor difference in the peptide response between these assays. This might be due to the different populations to be identified by assays; the library assay identified IFNγ+CD8+CD45RO+ cells whereas AIM assay identified the cells which became positive for selected activation markers after peptide stimulation.

Next, we further expanded the libraries obtained from moderate convalescents and performed the following downstream experiments because the libraries from severe convalescents tended not to expand well and did not reach enough cells. The effector features of $M_{198–206}$-specific CD8+ T cells, cytokine profiles were examined by intracellular cytokine staining (ICS). As shown in Fig. 3a, $M_{198–206}$-specific CD8+ T cells produced the typical inflammatory cytokines: IFNγ and TNFα. Further, we examined cytotoxic activity by employing $M_{198–206}$ peptide-pulsed Calu-3 cells as a model of SARS-CoV-2-infected lung epithelial cells. $M_{198–206}$-specific CD8+ T cell lines (Supplementary Fig. 6) from the

libraries were subjected to the assay. As shown in Fig. 3b, antigen-specific cytotoxic activities were proportional to the effector/target (E/T) ratio. Furthermore, a TCR αβ pair cloned from an $M_{198-206}$-specific CD8+ T cell line was functional (Supplementary Table 2) (Supplementary Fig. 7). To examine direct effector functions of $M_{198-206}$-specific CD8+ T cells to SARS-CoV-2-infected lung epithelial cells, Calu-3 cells were infected with SARS-CoV-2-Wuhan strain for the coculture with $M_{198-206}$-specific CD8+ T cells. Remarkably, as shown in Fig. 3c, $M_{198-206}$-specific CD8+ T cells suppressed not only intracellular viral RNA replication, but also suppressed propagation of infectious virus in SARS-CoV-2-infected Calu-3 cells, identifying $M_{198-206}$ as a SARS-CoV-2 epitope of CTLs. To examine immunological relevance of $M_{198-206}$-specific CD8+ T cells, we

studied the $M_{198-206}$-specific CD8+ T cells in the peripheral blood of COVID-19 convalescents with different clinical severities without in vitro expansion. As expected, tetramer-positive CD8+ T cells were detected (Fig. 4a, b) with significantly higher frequency in peripheral blood of moderate or severe COVID-19 convalescents, which is consistent with the results from the libraries (Fig. 2d). In addition, we examined the frequency of virus-specific CD8+ T cells which were reported as prevalent virus-specific CD8+ T cells in previous reports[30,32,33]. As shown in Supplementary Fig. 8d, in our cohort, the frequency of $M_{198-206}$-specific CD8+ T was higher than that of ORF1b or ORF3a-specific CD8+ T cells. Further, additional convalescents who suffered moderate/severe COVID-19 in early 2022 (CV-052, 057, 062, 065, 071, and 073) (Supplementary Table 1), when Omicron strain spread in Japan, were examined. In fact, among the six convalescents from moderate/severe COVID-19, three were tested for Omicron and all of them were found to be positive for Omicron. Five out of six moderate-severe COVID-19 convalescents harbored $M_{198-206}$-specific CD8+ T cells with similar frequency to late 2020 (Fig. 4g). Importantly, $M_{198-206}$ specific CD8+ T cells were detected in the peripheral blood of COVID-19 convalescents for more than a year (Fig. 4f). We also found that an $M_{198-206}$-specific CTL line suppressed propagation of Omicron strain (Supplementary Fig. 8c). Taken together, we concluded that $M_{198-206}$ is an immunoprevalent CTL epitope in our study cohort.

## Phenotypes and signatures of $M_{198-206}$-specific CTLs

By using $M_{198-206}$ MHC tetramer, we addressed the question as to whether the status of the virus-specific CD8+ T cells, such as

### Table 1 | The candidates of SARS-CoV-2-M epitopes for HLA-A*24:02

| Start | End | Length | Peptide | Score | Percentile rank |
|-------|-----|--------|-----------|----------|-----------------|
| 95 | 103 | 9 | YFIASFRLF | 0.913594 | 0.02 |
| 94 | 102 | 9 | SYFIASFRL | 0.83484 | 0.05 |
| 46 | 54 | 9 | LYIIKLIFL | 0.685153 | 0.11 |
| 198 | 206 | 9 | RYRIGNYKL | 0.670153 | 0.12 |
| 54 | 62 | 9 | LWLLWPVTL | 0.522979 | 0.18 |

Full-length amino-acid sequence of M protein was subjected to in silico analysis of epitope screening using MHC-I-binding prediction tool (v2.24) on immune epitope database (see Methods). Top five candidate peptides are shown.

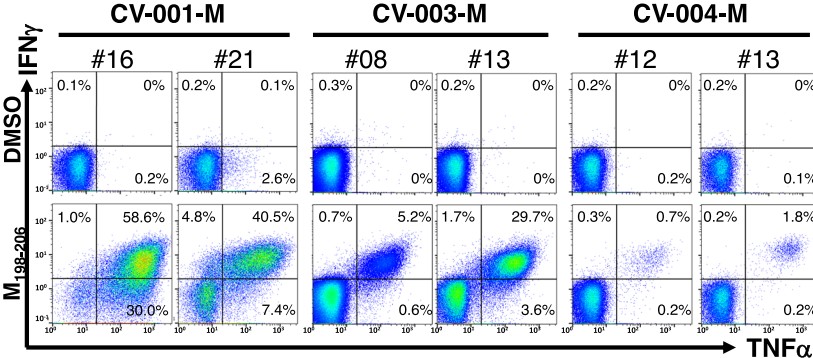

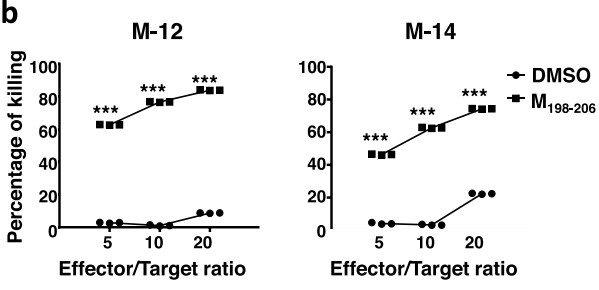

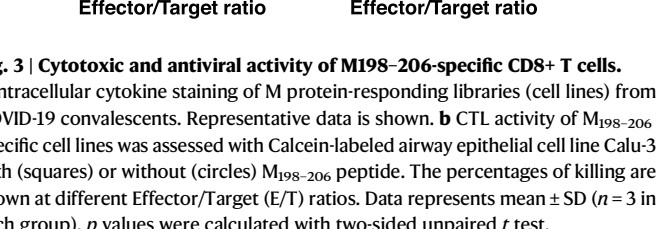

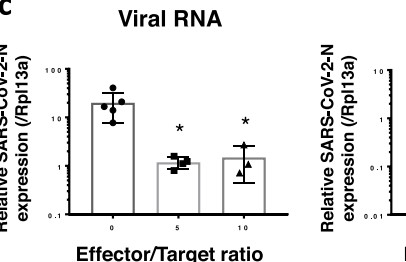

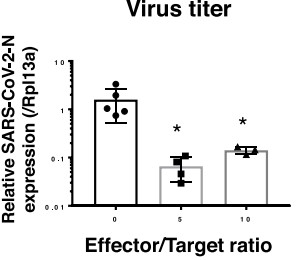

**Fig. 3 | Cytotoxic and antiviral activity of M198–206-specific CD8+ T cells.**
**a** Intracellular cytokine staining of M protein-responding libraries (cell lines) from COVID-19 convalescents. Representative data is shown. **b** CTL activity of $M_{198-206}$ specific cell lines was assessed with Calcein-labeled airway epithelial cell line Calu-3 with (squares) or without (circles) $M_{198-206}$ peptide. The percentages of killing are shown at different Effector/Target (E/T) ratios. Data represents mean ± SD ($n = 3$ in each group). p values were calculated with two-sided unpaired t test.
***$p = 3.3 \times 10^{-10}$, $4.2 \times 10^{-10}$, and $7.1 \times 10^{-11}$ for M-12; ***$p = 2.0 \times 10^{-8}$, $1.4 \times 10^{-9}$, and $1.2 \times 10^{-9}$ for M-14. **c** Calu-3 cells were infected with SARS-CoV-2 at an MOI of 0.1. CTLs were added to the cells with the indicated effector (CTL)/target (Calu-3) ratio

at 24 h after infection. After a further 24 h incubation, cells were washed and harvested. Intracellular viral RNA was measured by quantitative real-time PCR using a primer pair targeting the SARS-CoV-2 N gene region. The titer of infectious virus in the supernatant was measured based on the amount of intracellular viral RNA in VeroE6/TMPRSS2 cells infected with virus-containing supernatant from Calu-3 cells. The expression level of viral RNA was normalized to host *rpl13a* expression. Data represent mean ± SD ($n = 5$, 4, and 3 for E/T = 0, 5, and 10, individually). p values were calculated by two-sided Mann–Whitney test. *$p = 0.016$, 0.036 for viral RNA; *$p = 0.016$, 0.036 for virus titer.

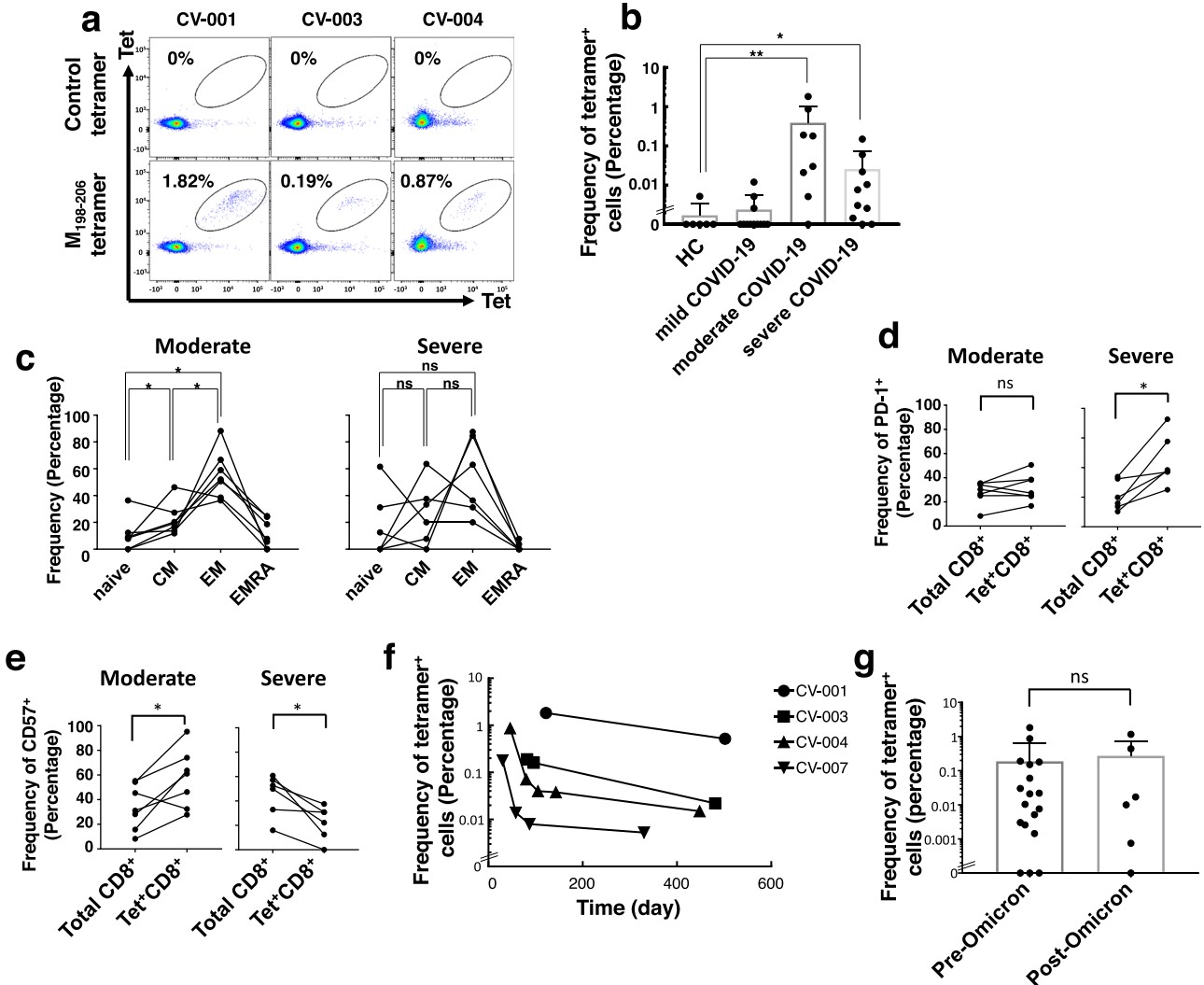

**Fig. 4 | Phenotypic analysis of M198−206-specific CD8+ T cells in PBMC.** Flow cytometric analysis of $M_{198−206}$-specific CD8$^+$ T cells. **a** Representative data of $M_{198−206}$ tetramer staining of PBMC from moderate COVID-19 convalescents. **b** The frequencies of tetramer-positive cells were compared among indicated groups (healthy donors (HC), $n = 6$; mild, $n = 12$; moderate, $n = 8$; severe, $n = 10$). Data represent mean + SD. $p$ values were calculated by a two-sided Mann–Whitney test. **$p = 0.0073$, *$p = 0.024$. **c** Subsets of $M_{198−206}$-specific CD8$^+$ T cells from COVID-19 convalescents ($n = 13$; 7 moderate and 6 severe subjects) were analyzed based on CD45RA and CCR7 staining. $p$ values were calculated by two-sided Wilcoxon matched-pairs signed rank test. *$p = 0.047$ (naive-CM), 0.031 (CM-EM), and 0.031 (naive-EM) for moderate; *$p = 0.69$ (naive-CM), 0.31 (CM-EM), and 0.19 (naive-EM)

for severe. **d**, **e** The frequency of PD-1 (**d**) and CD57 (**e**) positive cells were compared between total CD8$^+$ T cells (Total CD8$^+$) and $M_{198−206}$-MHC tetramer$^+$ CD8$^+$ T cells (Tet$^+$ CD8$^+$) in moderate ($n = 7$) and severe ($n = 6$) groups. $p$ values were calculated by two-sided Wilcoxon matched-pairs signed rank test. ns = not significant, *$p = 0.031$ for (**d**); *$p = 0.031$, 0.031 for **e**. **f** Time-course study of the frequency of $M_{198−206}$-specific CD8$^+$ T cells in COVID-19 convalescents. **g** The frequency of $M_{198−206}$ tetramer-positive cells among moderate/severe COVID-19 convalescents of pre-SARS-CoV-2-Omicron and post-SARS-CoV-2-Omicron pandemic. $p$ values were calculated by two-sided Mann–Whitney test. Data represent mean + SD ($n = 18$, and 6 for Pre-Omicron and Post-Omicron, individually).

differentiation, exhaustion, or senescence, reflects disease severity. With flow cytometry we found that the tetramer-positive CD8$^+$ T cells in peripheral blood were significantly skewed toward an effector-memory (CCR7$^-$ CD45RA$^-$) phenotype in moderate group, but not in severe group (Fig. 4c). The frequency of inhibitory receptor PD-1$^+$ cells in tetramer-positive CD8$^+$ T cells was higher than that in total CD8$^+$ T cells in the severe group, suggesting increased exhaustion of $M_{198−206}$-specific CTLs; this observation supports several recent reports (Fig. 4d)[34–36]. Interestingly, the senescence marker CD57$^+$ was significantly higher in tetramer-positive CD8$^+$ T cells of patients in the moderate group, but not those of patients in the severe group (Fig. 4e). These observations were not due to the difference in the time point after the infection (Supplementary Fig. 8a). The gating strategy for these analyses is demonstrated in Supplementary Fig. 9. Functionality of $M_{198−206}$-specific CD8$^+$ T cells in the moderate group was confirmed

by detection of response to peptide stimulation; they secreted IFNγ and TNFα upon the peptide stimulation as observed in the libraries (Supplementary Fig. 8b).

Next, we performed single-cell RNA-sequencing (scRNA-seq, 10X Genomics Platform) analysis on $M_{198−206}$-specific CD8$^+$ T cells. A total of 18,222 tetramer-positive CD8$^+$ T cells were isolated via florescence-activated cell sorting (FACS) from 10 PBMC samples derived from 6 convalescents (three moderate and three severe convalescents; in two moderate convalescents, samples from different time points after disease onset were included (Supplementary Table 3). Each sample was individually stained with Hashtag antibodies, followed by FACS-based isolation, then mixed and subjected to sequencing. As a result, single-cell transcriptomic data were obtained from 4,452 single cells. Uniform manifold approximation and projection (UMAP), a bioinformatic dimension reduction algorithm, identified 11 clusters (cluster

0 to 10) (Fig. 5a). Since cluster 9 (C9) and C10 did not include enough number of cells (<1%), these clusters were removed from the further analysis. Differentially expressed genes (DEGs, one cluster vs. rest of the cells) of each cluster and selected featured genes are shown in Fig. 5b. Clusters could be roughly divided into two groups (group 1 and 2); group 1 includes C1, C2, C3, C4 and C7 and group 2 includes C0, C5, C6, and C8. Of note, group 1 highly expressed cytotoxic-effector genes including *GZMB*, *GZMA*, and *TBX21* with some preferences in the expression of *FCGR3A*, *PRF1*, *GNLY*, *CX3CR1*, *GZMH*, or activation markers *HLA-DR* and *CD38* etc. (Fig. 5c and Supplementary Fig. 10a). Additionally, Gene set enrichment analysis with a consensus list of cytotoxicity signature genes[37], also showed high score in group 1 (Fig. 5g), demonstrating that these are the cytotoxic-effector or memory cells ($T_{cyto-eff/mem}$). *B3GAT1 (CD57)* expression was also localized in group 1 clusters (Supplementary Fig. 10a). Furthermore, DEG analysis between group 1 and 2 showed that in group 1, several cytotoxicity markers were upregulated while some of the naïve markers (e.g., *CCR7*, *TCF7*, *LEF1*) were downregulated, demonstrating that the group 2 includes naïve/less differentiated cells (Fig. 5d–f, Supplementary Fig. 10a). In fact, naive markers including *CCR7*, *CD62L*, *CD28*, and *CD27* were highly expressed in C8, suggesting that C8 is the cluster of naïve cells or memory stem cells (hereafter designated as 'T$_{naive-like}$') (Fig. 5c, d and Supplementary Fig. 10a). The expression of naive markers was gradually decreased from C8 towards C5 (Fig. 5d), indicating the early differentiated status of C5. To elucidate the features distinguishing C0 and C6 from the rest of clusters, we explored marker genes (see "Methods"). Curiously, *GZMK* was identified as a selectively expressed gene in both C0 and C6, whose expression was slightly decreased toward C6 (Fig. 5c, d). Recent papers have reported *GZMK* as a marker of predysfunctional cells or precursor of exhausted T($T_{PEX}$) cells, which have a distinct fate commitment to exhausted cells ($T_{EX}$) in human memory T cell pool[38]. In line with this, C0 highly expressed *TCF7* and *IL7R* and intermediately expressed *ZNF683*, *PDCD1* (Fig. 5d, Supplementary Fig. 10a) consistent with gene signature of $T_{PEX}$ cells. As expected, C6 highly expressed $T_{EX}$ markers including inhibitory receptors such as *PDCD1* and *TIGIT* (Fig. 5d). Regarding other inhibitory receptors, *CD244* was expressed in C6 ($T_{EX}$) as previously reported[39] (Supplementary Fig. 10a). Another inhibitory receptor *LAG3* was expressed in C6 ($T_{EX}$)and C0 ($T_{PEX}$)[40,41] (Supplementary Fig. 10a). Cytotoxic genes *PRF1* and *GZMB* were highly expressed in C6 and moderately in C0, that are similar to the characters of $T_{EX}$ and $T_{PEX}$ cells[39,41] (Fig. 5d and Supplementary Fig. 10a). The expression pattern of transcription factor genes such as *EOMES*, *TOX*, *TCF7*, *TBX21*, are also consistent with previously reported signatures of $T_{EX}$ and $T_{PEX}$ cells[41–47] (Fig. 5d and Supplementary Fig. 10a). These findings could support that C6 and C0 are corresponding to $T_{EX}$ and $T_{PEX}$, respectively. Moreover, C6 showed high score by analysis with exhaustion signature gene list employed in a recent work[37] (Fig. 5h). Trajectory analysis revealed the sequential distribution of the heterogenous cell-states along the C0-C6 axis towards C6 (Fig. 5i). Thus, we concluded that trajectory from C0 to C6 represents a unique trajectory towards the exhaustion of SARS-CoV-2-specific CD8⁺ T cells. Although the number of convalescent subjects was small, cells from subjects with moderate disease tended to group into $T_{cyto-eff/mem}$ cells (group 1 clusters). In contrast, cells from subjects in the severe category tended to group into $T_{EX}$ cells (C6) (Fig. 5j). This observation supported the results from the flow cytometry analysis where the severe group had higher PD-1 and lower CD57 expression (Fig. 4d, e). Furthermore, to confirm the increased exhaustion phenotype of M$_{198–206}$-specific CD8⁺ T cells in the severe group, we focused on *TIGIT*, an additional exhaustion marker for T cell, because C6 specifically co-expressed *PDCD1* and *TIGIT* (Supplementary Fig. 10b). We compared the frequency of PD-1⁺TIGIT⁺ cells between moderate and severe group through flow cytometry analysis of SARS-CoV-2-M$_{198–206}$-specific CD8⁺ T cells. As expected by scRNA-seq results, the frequency of the cells was significantly higher in the cells from severe group compared with those from moderate group (Fig. 5k). Of note, time post viral clearance was not correlated with the frequency of PD-1⁺TIGIT⁺ cells (Supplementary Fig. 8e).

## Identification of public TCRs of virus-specific CD8⁺ T cells

scRNA-seq analysis provided TCR sequences of the SARS-CoV-2-M$_{198–206}$ specific CD8⁺ T cells. TCR clonal expansion analysis revealed that in all the clusters except C8 (T$_{naive-like}$) and C5, a majority of the clones expanded well (more than three cells observed, Fig. 6a) as expected by the signature of those clusters (T$_{cyto-eff/mem}$, T$_{PEX}$, and T$_{EX}$). Among TCR sequences of the top 20 expanded clones, clone 14 from the CV-001 convalescent had the same sequences as TCRα$_{rank1}$β$_{rank1}$ clone isolated from CD8⁺ T cell libraries (Supplementary Table 2, Supplementary Table 5). Interestingly, clone 58 from the same convalescent had the same amino acid sequences of α and β chains as clone 14 while there is a difference of a single nucleotide of α chain, suggesting that this TCR was advantageous in forming significant proportions of CTLs in this convalescent. Longitudinal analysis of the top 20 expanded clones from two convalescents (CV-001 and CV-004) detected long-lived memory CD8⁺ T cells resided likely in C0, C2, and C4, suggesting that these clusters included memory cells. (Fig. 6c).

Although obtained TCR sequences showed the heterogeneity of the TCR clones, there were a few TCRs shared among different convalescents; they are demonstrated as common TCRβ-1, -2, -3, and -4 in Fig. 6b and Table 2. Of note, amino acid sequence of CDR3 in common TCRβ-1 was shared among all members of the moderate group of convalescents (CV-001, CV-003, and CV-004) with difference in a single nucleotide in all the subjects. Among TCR clones with common TCRβ−1, clone 151 and clone 893 had identical amino acid sequence of CDR3 in TCRα while clone 91 had similar but different amino acid sequence of CDR3 in TCRα, in which an amino acid at position 4 of CDR3 was different (Table 2). Moreover, amino acid sequence of CDR3 in TCRβ-2 was different only in an amino acid at position 8 from TCRβ-1. Among TCR clones with TCRβ-1 or TCRβ-2, clone 29, clone 151 and clone 893 had identical amino acid sequence of CDR3 in TCRα while clone 209 had very closed, but different amino acid sequence of CDR3 in TCRα, in which an amino acid at position 4 of CDR3 was different from clone 151, 893, 91, and 29. Collectively, we found public TCRαβ motif: "CAVXYNQGGKLIF" for α motif and "CASSDSGXDGYTF" for β motif. The identification of the public TCRs also could highlight the importance of M$_{198–206}$ immunoprevalent epitope recognition in SARS-CoV-2 clearance.

## Discussion

Even at present, SARS-CoV-2 is spreading across the world and accumulating mutations and causing COVID-19 with diverse clinical features. Especially in elder populations, life-threating outcomes are manifested; age-specific infection fatality rate has been estimated to exponentially increase to 15% at age 85[1]. Intensive efforts to tackle this issue are on-going from immunological point of view. Recent comprehensive analyses demonstrated innate immunity (e.g., type I interferon) is a critical contributing factor to the course of COVID-19. In parallel with innate immunity, contributions of the adaptive immune system (e.g., cellular immunity) have been demonstrated[3,5]. In this study, we focused on CTLs, a critical effector population for virus clearance, and performed a thorough characterization of virus-specific CTLs from COVID-19 convalescents with different severities focused on M$_{198–206}$, an immunoprevalent CTL epitope to understand how differentiation status of virus-specific CTLs is related to disease severity.

We screened T cell libraries as previously reported[18,48–50] by constructing several artificial antigen-presenting cells which express SARS-CoV-2 proteins, individually. The system was optimized to screen for

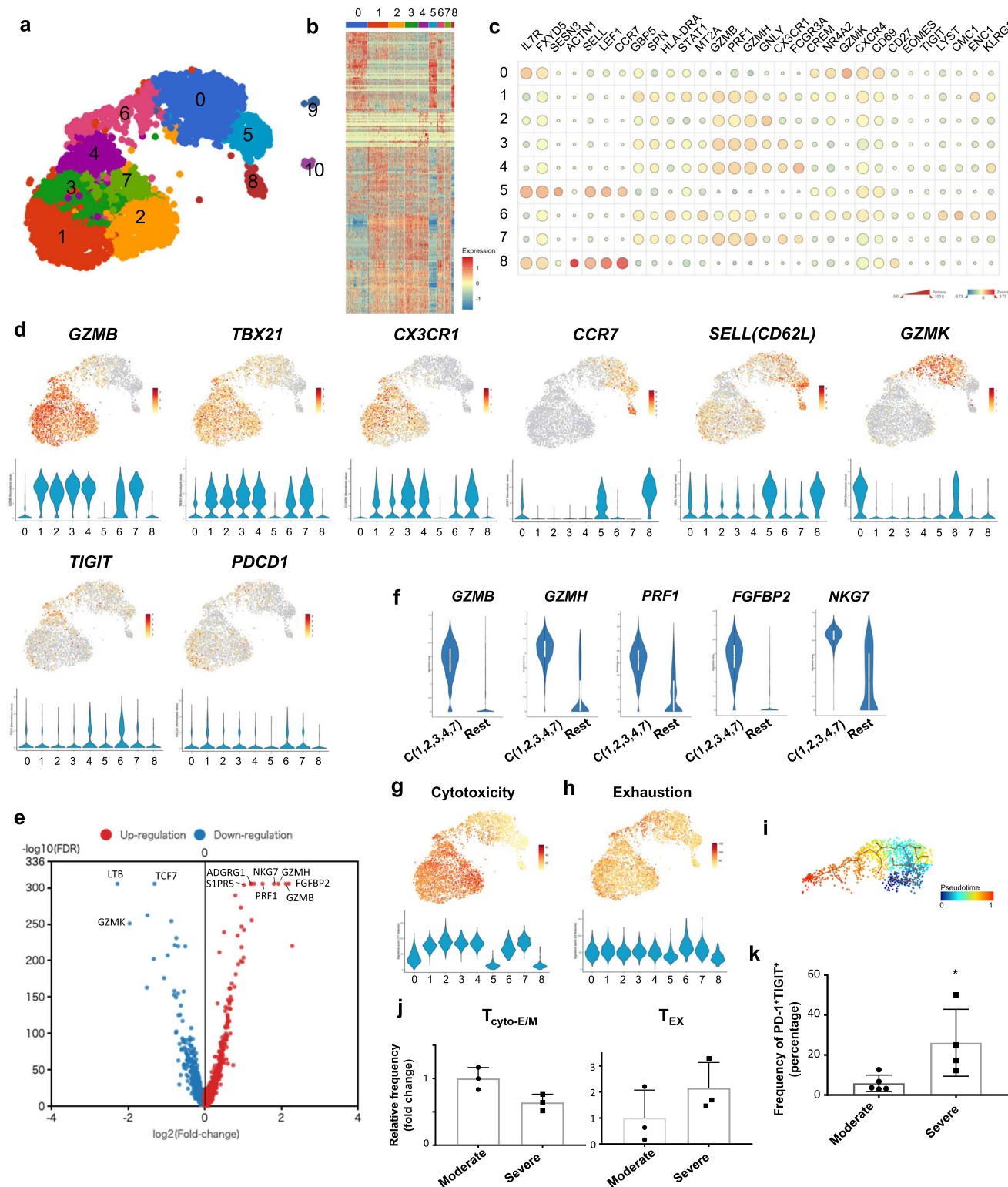

**Fig. 5 | Single-cell RNA-sequencing of M198–206-specific CD8+ T cells. a** UMAP analysis displays transcriptomic landscape of M$_{198-206}$-specific CD8$^+$ T cells. Each dot corresponds a single cell and is colored according to the cluster. **b** Heatmap representation of top 50 differentially expressed genes of each cluster (each cluster vs. rest). **c** Mean expression (color) and frequency of expressing cells (size) of the featured genes were shown in the graph. **d** UMAPs are illustrating mRNA expression of the indicated genes. The expression level of the indicated genes of each cluster was shown in the violin plot. **e** Volcano plot shows fold changes of differentially expressed genes between cluster (1, 2, 3, 4, 7) and the rest. **f** Violin plot shows mRNA expression of indicated genes in cluster (1, 2, 3, 4, 7) (n = 2727 cells) and the rest

(n = 1656 cells). Box represents 25–75 percentiles and median (straight line) and mean (dotted line) are indicated. **g, h** UMAPs indicate cytotoxicity (**g**) and exhaustion (**h**) signature score (see Methods). **i** Trajectory analysis shows transition of the cell state of indicated cells within cluster 0 and cluster 6. **j** Graph shows relative frequency of T$_{cyto-eff/mem}$ and T$_{EX}$ cells from moderate (n = 3) or severe convalescents (n = 3). Data represent mean ± SD. **k** Frequency of PD-1$^+$TIGIT$^+$ cells from moderate (n = 5) and severe (n = 4) convalescents in M$_{198-206}$-tetramer$^+$ CD8$^+$ T cells were examined by flow cytometry analysis. Data represent mean ± SD. p values were calculated by two-sided Mann–Whitney test. *p = 0.032.

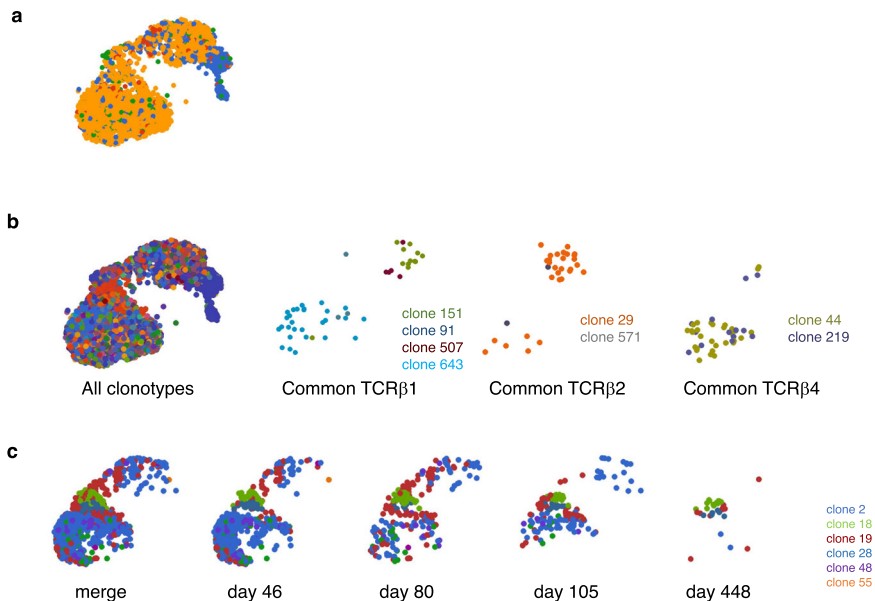

**Fig. 6 | Single-cell TCR-sequencing of M₁₉₈₋₂₀₆-specific CD8+ T cells. a** UMAP shows clonal expansion of M$_{198-206}$-specific CD8$^+$ T cells. Each color indicate the number of cells observed for each clone. Yellow, three or more cells; red, two cells; blue, single-cell; green, others. The cells from time-course study samples (Hashtag 2, 3, 4, and 6, see Supplementary Table 3) were omitted from the analysis. **b** Each T cell clone having common TCRβ1, TCRβ2, and TCRβ4 were shown in UMAPs. Colors represent different clones as indicated. **c** Longitudinal analysis of T cell clones of a moderate COVID-19 convalescent, CV-001 from the list of Top20 clones were shown in different colors as indicated on UMAP.

virus-specific IFNγ$^+$ CD8$^+$ T cells. M protein was identified as an immunocompetent viral protein, and M$_{198-206}$ was the dominant epitope of the M protein. Compared to the recent papers reporting comprehensive analyses of peptide epitopes in which many structural and non-structural protein epitopes were identified[22,51], our screening results were skewed toward the M protein. This discrepancy could come from the difference in the epitope preference of each HLA haplotype since our study focused on HLA-A$^*$24:02. Additionally, the other papers identified cross-reactive T cells present prior to the COVID-19 pandemic in their cohorts[52,53], which we could not detect in our cohort. It could also influence the memory T cell pool in the convalescents and could affect the screening results, accordingly. The other possibility was that the polyclonal expansion step in our system, driven by TCR stimulation and cytokines etc. could favor specific memory or effector T cell clones because theoretically there should be T cell subpopulations having different proliferative capacities within the population of CD45RO$^+$ cells. Thus, we might have underestimated the antigen-specific responses. On the other hand, one of the strong points of the assay is the ease of downstream analysis such as cytotoxicity assays, cytokine profiling by ICS, and tetramer staining, etc. The recently developed activation-induced marker (AIM) assay and ELISpot assay have successfully identified antigen-specific cells and are employed by many researchers. In a simple way, which increases the accessibility and feasibility of the assays, the assays picked the antigen-specific cells primed by the epitope in vivo. But since these assays need to consume most of subjected cells for analysis, it is difficult to perform the downstream analysis, which requires sufficient numbers of cells. Above all, it should be noted that all these assays focus on different aspects of the cells; for example, the AIM assay focuses on activation marker-positive cells shortly after the peptide stimulation, whereas the library assay focuses on IFNγ-producing CD8$^+$ T cells from CD45RO$^+$ population in this study.

Importantly, M$_{198-206}$-specific CD8$^+$ T cells were detected with significantly higher frequency at the periphery of convalescents suffering moderate/severe COVID-19 from late 2020 through early 2022, consistent with the reports demonstrating M protein as well conserved viral protein among various SARS-CoV-2 strains (https://nextstrain. org/ncov/gisaid/global). In addition, the M$_{198-206}$-specific CD8$^+$ T cell lines showed antigen-dependent IFNγ and TNFα production as well as cytotoxic activity and suppressed propagation of SARS-CoV-2 strains: Wuhan and Omicron. Thus, we concluded that M$_{198-206}$ was one of the immunoprevalent CTL epitopes in our cohort. Previously M$_{198-206}$ was examined as an HLA-A*30:01 or HLA-A*24:02-restricted CD8$^+$ T cell epitope, but has not been highlighted as a crucial viral epitope associated with COVID-19 severity[29,33,54].

Next, we performed in-depth analysis of the immunoprevalent epitope M$_{198-206}$-specific CD8$^+$ T cells to examine their characteristics. M$_{198-206}$-specific CD8$^+$ T cells were detected at substantially high frequency in the moderate and severe groups, which enabled us to study their phenotypic differences by conventional flow cytometry analysis. We found that the exhaustion marker PD-1 was significantly high in the severe group, compared with the moderate group. In contrast, senescence/terminal differentiation marker CD57 was significantly lower in the severe group than the in moderate group. Additionally, tetramer-positive cells of the moderate group were highly skewed towards effector-memory cells, which was disturbed in the cells of the severe group.

We further took advantage of the high frequency of M$_{198-206}$-specific cells in the moderate and severe groups of our cohort and performed scRNA-seq analysis by isolating a total of 18,222 tetramer-positive cells from moderate and severe convalescents by FACS. As a result, this revealed a highly heterogenous state of the virus-specific CD8$^+$ T cells. In addition to cytotoxic-effector/memory populations, we found exhausted/pre-exhausted populations along a unique trajectory; the trajectory was from *GZMK*-expressing progenitors of exhaustion/pre-exhausted cells to related exhausted cells, which had the highest signature scored by gene set enrichment analysis with inhibitory receptor expression such as *PDCD1* and *TIGIT* and limited cytotoxic gene expression. Surprisingly, T$_{PEX}$ cluster accounted for the biggest population on UMAP. It is of note that the cells from severe group tended to be accumulated into T$_{EX}$ cell cluster, compared with those from the moderate group, which supported the results from the PD-1 and/or TIGIT staining experiment by flow cytometry analysis (Figs. 4c−e, 5k). As reported in chronic viral infection or cancer[55],

**Table 2 | Identification of Public TCRs**

| common TCRβ | clone ID | Hashtag | sub ID | TCRβ amino acid | TCRβ V | TCRβ J | TCRβ C | TCRα amino acid | TCRα V | TCRα J | TCRα C |
|---|---|---|---|---|---|---|---|---|---|---|---|
| 1 | 151 | 5, 6 | CV-001 | **CASSDSGADGYTF** | TRBV6-4 | TRBJ1-2 | TRBC1 | **CAVIYNQGGKLIF** | TRAV1-2 | TRAJ23 | TRAC |
| | (507) | 5, 6 | CV-001 | **CASSDSGADGYTF** | TRBV6-4 | TRBJ1-2 | TRBC1 | | | | |
| | 893 | 7 | CV-003 | **CASSDSGADGYTF** | TRBV6-4 | TRBJ1-2 | TRBC1 | **CAVIYNQGGKLIF** | TRAV1-2 | TRAJ23 | TRAC |
| | 91 | 1, 2 | CV-004 | **CASSDSGADGYTF** | TRBV6-4 | TRBJ1-2 | TRBC1 | **CAVYNQGGKLIF** | TRAV1-2 | TRAJ23 | TRAC |
| | (643) | 1 | CV-004 | **CASSDSGADGYTF** | TRBV6-4 | TRBJ1-2 | TRBC1 | | | | |
| 2 | 29 | 5, 6 | CV-001 | **CASSDSGTDGYTF** | TRBV6-4 | TRBJ1-2 | TRBC1 | **CAVIYNQGGKLIF** | TRAV1-2 | TRAJ23 | TRAC |
| | (571) | 5 | CV-001 | **CASSDSGTDGYTF** | TRBV6-4 | TRBJ1-2 | TRBC1 | | | | |
| | 209 | 7 | CV-003 | **CASSDSGTDGYTF** | TRBV6-4 | TRBJ1-2 | TRBC1 | **CAVLYNQGGKLIF** | TRAV1-2 | TRAJ23 | TRAC |
| 3 | 13 | 1 | CV-004 | CASRGAVDTQYF | TRBV19 | TRBJ2-3 | TRBC2 | CVVPLGGSNYKLTF | TRAV12-1 | TRAJ53 | TRAC |
| | 293 | 6 | CV-001 | CASRGAVDTQYF | TRBV19 | TRBJ2-3 | TRBC2 | CAALDNDMRF | TRAV13-1 | TRAJ43 | TRAC |
| 4 | 44 | 7 | CV-003 | CASSQEGIEQFF | TRBV3-1 | TRBJ2-1 | TRBC2 | CAGVSSNTGKLIF | TRAV8-3 | TRAJ37 | TRAC |
| | 219 | 5 | CV-001 | CASSQEGIEQFF | TRBV3-1 | TRBJ2-1 | TRBC2 | CAETGTASKLTF | TRAV5 | TRAJ44 | TRAC |

Common TCRs found in two or more subjects were summarized. CDR3 sequences, as well as V, J (and C) usage were shown for each clone. The sequences of common TCRβ-1 and paired TCRα-1 and paired TCRα chains were shown in bold, most of which are shared with common TCRβ-2 and paired α chains. ID numbers with brackets indicate the clones having no TCRα information.

functional loss of exhausted SARS-CoV-2-specific CTLs could result in a failure of proper elimination of the virus, which results in severe outcomes. There has been a debate regarding the exhaustion state of T cells in COVID-19. Recently Dr. Shin's group beautifully showed that PD-1 expressing CD8[+] T cells in COVID-19 were functionally active in terms of IFNγ production[56,57]. It would not be contradictory because the loss of IFNγ production occurs only in severely exhausted PD-1-expressing CD8[+] T cells as reviewed in elsewhere[57,58].

There are several papers performing scRNA-seq on SARS-CoV-2-specific CD8[+] T cells[37,59]. Early studies successfully analyzed AIM[+] cells as SARS-CoV-2-specific CD8[+] T cells[37], but MHC tetramer technology would capture different status of the cells because this could isolate the cells without stimulation. Recently an excellent study was reported by Dr Dong's group; NP$_{105-113}$-B*07:02-sepecific CD8[+] T cells were extensively analyzed with MHC-NP$_{105-113}$ peptide tetramer[51,59]. NP$_{105-113}$-specific CD8[+] T cells were frequently detected in mild group, but less in the severe group, and clonotypes with different functional avidity were detected in their cohort. Consistent with our results, scRNA-seq analysis showed high expression of granzyme K in NP$_{105-113}$-specific CD8[+] T cell from severe group[59]. Thus, together with this report, our data strengthened the basic concept that CTLs are as crucial immune effector cells to determine disease severity.

Single-cell TCR-sequencing analysis identified numerous clonotypes among the COVID-19 convalescents. Of note, several public TCRs were identified. These included public TCRαβ-motifs shared in 3/3 of the moderate convalescents, which could further highlight the importance of the M$_{198-206}$ epitope recognition in COVID-19. As expected, the transcriptomic status of the cells (i.e., distribution of the cells on UMAP) within and across the clonotypes were heterogenous, suggesting that the fate of CTLs such as cell division, differentiation, survival or cell death should have been regulated by various factors including basal state of the cells, amount and frequency of the antigen stimulation or co-stimulation, cytokines etc., during T cell activation in the clinical course of disease. Recent advanced platforms are accumulating information on public TCRs in infectious diseases and cancers[60–62]. In HIV-infected patients, there are rare subjects who could control viral propagation without therapy, called HIV controllers[61]. Public TCRs were reported to be crucial for control of HIV in those subjects; these TCRs showed high affinity to Gag293, which the most immunoprevalent CD4 epitope in HIV capsid[61]. Although further analyses are required, public TCRs identified in our experiment might contribute to recovery or lead to less severity. Since public clonotypes of SARS-CoV-2-specific T cells has been accumulating[63,64], our information on public clonotypes of M-specific CTL could be important to consider further strategies against unresolved disaster.

We analyzed CTL phenotype in the context of a single M$_{198-206}$ epitope without including other epitopes or bystander T cells as controls for scRNA-seq analysis. Also, these analyzes were performed with low numbers of moderate/severe COVID-19 convalescents. Additionally, one HLA haplotype: HLA-A*24:02 was focused on in this study because of high frequency in our study cohort. These are limitations of this study, therefore, continuous accumulation of the data from different cohorts with different immunocompetent epitopes is required. Furthermore, deeper association of exhausted phenotype and signature with SARS-CoV-2-M$_{198-206}$-specific CTLs from convalescents of severe disease could raise the possibility; it is a consequence of heightened immune activation that is associated with severer disease. However, since we observed less coordinated differentiation status of SARS-CoV-2-M$_{198-206}$-specific CTLs in the severe group comparing to the moderate group (Fig. 4c), it could be interpreted as a cause of severe COVID-19. In order to corroborate such an interpretation, detailed investigation of virus-specific CTLs in immune compromised hosts with different severity are required.

In conclusion, we propose the trajectory towards exhaustion as a SARS-CoV-2-specific CTL fate to dysfunction in COVID-19. This could

lead to poor outcomes presumably due to insufficient innate immune system in COVID-19 (e.g., type 1 IFN signaling etc.). Moreover, $M_{198-206}$ could be highlighted as a crucial CTL epitope to determine COVID-19 severity. These results could provide a platform for understanding severe COVID-19 pathogenesis in relation with dysfunction of cellular immunity.

## Methods

### Study participants and ethics

COVID-19 convalescents were recruited from hospitals affiliated with Hyogo Medical University or Kyowakai Medical Corporation. COVID-19 convalescents were classified into three groups (i.e., mild, moderate, and severe) based on the extent of oxygen supplementation and requirement of mechanical ventilation (mild: no oxygen supplementation, moderate: oxygen supplementation $FiO_2 < 0.5$, severe: heavy oxygen supplementation $FiO_2 > 0.5$ and/or mechanical ventilation) according to a recent report[22]. COVID-19 RNA was detected by Ampdirect 2019.nCov2 (SHIMADZU) at Hyogo College of Medicine, one Step RT PCR at SRL or Aptima SARS-CoV-2 at Hokenkagaku-Nishinihon. COVID-19 antigen was detected by Lumipulse G SARS-CoV-2 Ag (Fujirebio Inc.) at Kyowakai Medical Corporation, ESPURAIN at SRL. Diagnosis of COVID-19 was performed based on the presence of either mRNA or antigen. SARS-CoV-2 antigen. Individuals without known significant health problems including suspected COVID-19 and without the COVID-19 vaccine treatment were enrolled as healthy volunteers. Participant information is shown in Supplementary Table 1. Ethical approval was given by the ethics committee of Hyogo College of Medicine (reference: 202104-144). The consent to publish clinical characteristics of all the participants was obtained through written informed consent. Peripheral blood was drawn from convalescents recovered from COVID-19 with different severities or healthy volunteers after written informed consent was given. PBMCs were isolated by Ficoll-Hypaque gradient centrifugation and genomic DNA were purified using QIAamp DNA blood mini kit (51104, Qiagen). All the subjects were tested for HLA-A DNA typing (GenoDive Pharma Inc.) and the ones positive for HLA-A*24:02 were subjected to the CD8+ T cell library assay and/or MHC tetramer staining etc. Our reporting of clinical data complies to the STROBE guidelines.

### Cell lines

VeroE6/TMPRSS2[65] (JCRB 1819) cells were at 37 °C in 5% $CO_2$ in Dulbecco's modified Eagle's medium (DMEM) (Thermo Fisher Scientific) supplemented with 10% heat-inactivated fetal bovine serum and 1 mg/ml G418. Calu-3 (ATCC HTB-55) cells, a human lung epithelial cell line, were maintained in Minimum Essential Medium (MEM) (Thermo Fisher Scientific) supplemented with 20% heat-inactivated fetal bovine serum. For the establishment of another VeroE6 cells expressing TMPRSS2, a vesicular stomatitis virus (VSV)-G pseudotyped lentivirus having human *tmprss2* gene was produced using 293FT cells. VeroE6 (ATCC) cells infected with the pseudotyped virus were selected with 300 mg/ml hygromycin for at least 1 week. These bulk-selected cells were used for detecting SARS-CoV-2 viral RNA in supernatants from infected Calu-3 cells. TG40/CD8a cells were cultured at 37 °C in 5% $CO_2$ in RPMI medium (Wako) supplemented with 10% heat-inactivated fetal bovine serum.

### Artificial antigen-presenting cell preparation

SARS-CoV-2 total RNA was provided by National Institute of Infectious Diseases. Viral genes including M, N, S, ORF3a, and ORF1ab nsp6 were cloned into pMSCVpuro (Clontech) plasmid DNA and transfected to retrovirus producing phoenix 293 T cells. For the artificial antigen-presenting cells, K562 cells were transfected with HLA-A*24:02:01 cDNA[66] (provided by RIKEN BRC DNA Bank) and costimulatory molecule 4-1BBL cDNA. Next the cells were transduced with the viral genes via retroviral overexpression system (Clontech) and antibiotic selection.

### Antibodies and reagents

The following monoclonal antibodies were used for flow cytometry analysis and cell sorting; anti-CD8a-Alexa Fluor 488 (300916, BioLegend, 1:200), anti-CD8a-FITC (FITC-65135, Proteintech, 1:50), anti-CD3-PE (317307, BioLegend, 1:200), anti-CD4-APC (300552, BioLegend, 1:200), anti-CD45RA-FITC (304148, BioLegend, 1:200), anti-CD45RO-PerCP/Cy5.5 (304222, BioLegend, 1:200), anti-PD-1-APC/Cy7 (329921, BioLegend, 1:50), anti-CD57-PerCP/Cy5.5 (359621, BioLegend, 1:200), anti-IFNγ-APC (502511, Biolegend, 1:200), anti-TNFα-BV421 (502931, BioLegend, 1:100), anti-CCR7-PE (353203, BioLegend, 1:50), anti-TIGIT-PerCP/Cy5.5 (372717, BioLegend, 1:50), anti-pan HLA (M0736, Dako, 1:200), control mouse IgG2a (401501, BioLegend, 1:1111), and anti-mouse IgG-FITC (406001, BioLegend, 1:200). LIVE/DEAD Fixable-Near IR Dead Cell Stain Kit (L10119, Thermo Fisher Scientific, 1:400), LIVE/DEAD Fixable-Aqua Dead Cell Stain Kit (L34957, Thermo Fisher Scientific, 1:400), and Human BD Fc Block (564220, BD Pharmingen, 1:50) was also used. MHC tetramer was prepared using QuickSwitch Quant HLA-A*24:02 Tetramer Kit-PE (TB-7302-K1, MBL Life Science) and QuickSwitch Quant HLA-A*24:02 Tetramer Kit-BV421 (TB-7302-K4, MBL Life Science) with appropriate peptides. CMV pp65$_{341-349}$ peptide (TS-0020-P, MBL) was purchased and all the other peptides were synthesized with a purity of >95% (Scrum Inc., or Toyobo).

### CD8+ T cell library assay

CD8+ T cell library assay was performed as previously described[18] with minor modifications. Briefly, CD8+CD45RO+ cells sorted from PBMC were cultured in 96-well round-bottom plates at $2 \times 10^3$ cells per well in complete DMEM medium supplemented with 5 mM HEPES, pH 7.3 (345-06681, Dojindo), 0.1 mM nonessential amino acids (139-15651, Fujifilm), 1 mM sodium pyruvate (190-14881, Fujifilm), 5% human serum (H3667, Sigma-Aldrich), 50 U/ml penicillin and 50 U/ml streptomycin (168-23191, Fujifilm) in the presence of 1 μg/ml PHA-L (11249738001, Roche) with 20 U/ml IL-2 (589104, BioLegend), 20 ng/ml IL-7 (581906, BioLegend), and 20 ng/ml IL-15 (570306, BioLegend) with $2 \times 10^4$ cells/well of irradiated allogeneic feeder cells. Fresh cytokines were added every 3 days. On day 9, the libraries were screened for antigen specificity by coculturing with irradiated-artificial antigen-presenting cells (aAPCs) expressing a series of viral proteins described above. CMV pp65$_{341-349}$ (QYDPVAALF) and Influenza PA$_{130-138}$ (YYLEKANKI) pulsed aAPCs (without expressing SARS-CoV-2 viral proteins) were also used. Then, the culture supernatant was harvested for IFNγ measurements by ELISA. In some experiments, CD45RO+CD8+ T cell libraries with positive responses were further expanded by adding cytokine cocktail every 3 days for approximately 14 days and then restimulated with control, SARS-CoV-2 viral protein-expressing aAPCs, or antigen peptide-pulsed aAPC.

### Flow cytometry and cell sorting

Flow cytometry were performed as previously described[18]. For cell-surface labeling, CD8+ T cell libraries or PBMCs were stained with the antibodies for 30 min on ice. The cells were then analyzed by BD LSRFortessa (BD Biosciences) with BD FACSDiva (V8.0) or MACSQuant Analyzer (Miltenyi Biotech) with MACSQuantify (version 2.4) or sorted using BD FACSAria II (BD Biosciences). Data analyses were performed with FlowJo (v10.4.2) (TreeStar). For the intracellular cytokine staining, cells were stimulated with indicated peptide for 2 hours and then further incubated in the presence of brefeldin A for 4 hours. After cell-surface staining, cells were fixed and permeabilized. Intracellular cytokines were detected with specific monoclonal antibodies using FoxP3/transcription factor staining buffer set (00-5523-00, eBioscience) according to the manufacturer's instructions.

## Real-time quantitative PCR

A GeneAmp 5700 sequence detection system (Applied Biosystems) and KAPA SYBR FAST (KK4621, Kapa Biosystems) were used to quantify the levels of indicated mRNA expressions. Thermal cycles was; 95 °C for 10 min, followed by 40 cycles of 95 °C for 15 sec and 60 °C for 1 min. The PCR primer sets used for real-time PCRs were as follows: S (5′-AACGCCACCAGATTTGCATC-3′,′-GTTTGCCCTGGAGCGATTTG-3′); M (5′-CGCGTTCCATGTGGTCATTC-3′, 5′-CCTTGATGTCACAGCGTC CT-3′); N (5′-CCTCGGCAAAAACGTACTGC-3′, 5′-TGGCACCTGTGTAGG TCAAC-3′); ORF3a (5′-CGGATGGCTTATTGTTGGCG-3′, 5′-GCAACGAG CAAAAGGTGTGA-3′); ORF1ab (5′-CTGCTAGTTGGGTGATGCGT-3′, 5′-AGCCCACATGGAAATGGCTT-3′); human *Actb* (5′-ACAGAGCCTCG CCTTTGC-3′ and 5′-CCACCATCACGCCCTGG-3′). In SARS-CoV-2 virus infection experiments, quantitative real-time RT-PCR was carried out using THUNDERBIRD™ SYBR qPCR Mix (TOYOBO) at 95 °C for 3 min, followed by 50 cycles of 95 °C for 10 sec and 60 °C for 1 min. Detection of fluorescence during the thermal cycling process and quantification studies were performed using CFX Connect™ Real-Time PCR detection system (BIO-RAD). The level of *rpl13a* expression in each sample was used to standardize the data.

The following primer sets were used: SARS-CoV-2 N (5′-AAATTTT GGGGACCAGGAAC-3′, 5′-TGGCAG CTGTGTAGGTCAAC-3′), human *rpl13* (5′-TGTTTGACGGCATCCCAC-3′, 5′-CTGTCACTGCCTGGTAC TTC-3′), and African green monkey *rpl13a* (5′-CTCAAGGTTGTG CGTCTGAA-3′, 5′-CTGTCACTGCCTGGTACTTCCA-3′).

## Western blot

Cells were harvested, lysed with lysis buffer (20 mM Tris-HCl; pH 7.5, 100 mM NaCl, 2 mM EDTA, 0.5% Triton X-100) supplemented with cOmplete protease inhibitor cocktail (11697498001, Roche). The lysates were electrophoresed on 8-20% SDS-polyacrylamide gels. Then samples were transferred to polyvinylidene difluoride membranes (Millipore), and blocked with 5% milk in Tris-buffered saline with 0.05% Tween-20 (TBS-Tween). Primary antibodies were as follows: anti-SARS-CoV-2-S antibody (GTX632604, GeneTex, 1/1000 in 5% milk TBS-Tween), anti-SARS-CoV-1-M antibody (AP6008b, Abgent, 1/1000 in 5% milk TBS-Tween), anti-SARS-CoV-2-N antibody (GTX135357, GeneTex, 1/1000 in 5% milk TBS-Tween), anti-SARS-CoV-2-ORF3a antibody (A20234, ABclonal, 1/1000 in 5% milk TBS-Tween), anti-SARS-CoV-2-NSP6 antibody (9177, ProSci, 1/1000 in 5% milk TBS-Tween), and anti-tubulin (T5168, Sigma-Aldrich, 1/10,000 in 5% milk TBS-Tween). Secondary antibodies were as follows: anti-mouse IgG-HRP (330, MBL, 1/10,000 in 5% milk TBS-Tween), anti-rabbit IgG (458, MBL, 1/10,000 in 5% milk). The signals were detected through the enhanced chemiluminescent substrate (WBKLS0100, Millipore, or 34094, Thermo Fisher Scientific) by Intelligent Dark BoxII (Fujifilm) using IR LAS-1000 Pro (version 2.5) software. Uncropped scans of the most important western blots are provided in the Source Data file.

## Cytotoxicity assay

$M_{198-206}$-specific CD8$^+$ T cells were enriched as effector cells as follows. $M_{198-206}$-responding libraries from COVID-19 convalescents were further expanded and enriched with cytokines in the presence of irradiated (45 Grey) M expressing aAPCs for a couple of weeks. Then the cells were harvested and $M_{198-206}$ specific cells were purified with $M_{198-206}$ MHC tetramer-PE and anti-PE microbeads (Miltenyi). The resulted cells were confirmed to be $M_{198-206}$ tetramer$^+$ with the purity of >95%. Next, in a 96-well plate, peptide-pulsed Calu-3 cells or unpulsed control cells were labeled with Calcein-AM (Dojin) as previously indicated[67]. Then, the effector cells were added or not added to the target cells with different E/T ratios as indicated. 24 hours later, the cells were extensively washed and intracellular calcein levels were measured using a fluorescence microplate reader Infinite M200 Pro (TECAN) ($\lambda_{Em}$ = 490 nm, $\lambda_{Ex}$ = 520 nm). Wells were triplicated for each condition and the percentage of killing was calculated as ((OD of no-effector added wells)-(OD of effector added wells))/(OD of no-effector added wells)×100.

## Preparation of SARS-CoV-2 virus stock

The SARS-CoV-2 isolate (UT-NCGM02/Human/2020/Tokyo)[68] and the Omicron isolate (BA.1 linage, TY38-873) from the National Institute of Infectious Diseases, Japan were propagated in VeroE6/TMPRSS2 (JCRB 1819) cells in DMEM containing 5% heat-inactivated fetal bovine serum at 37 °C in 5% CO$_2$. Briefly, SARS-CoV-2 was added at a multiplicity of infection (MOI) of 0.01 to VeroE6/TMPRSS2 (JCRB 1819) cells and incubated for 30 min at 37 °C. The culture medium was replaced with fresh medium. Cells were incubated for an additional 48 hours. The supernatant was centrifuged at 800×g for 5 minutes to remove cell debris. The supernatant was stored as virus stocks at −80 °C. The virus titer was determined by plaque assay using VeroE6/TMPRSS2 (JCRB 1819) cells.

## SARS-CoV-2 infection assay

Calu-3 cells were seeded at $2 \times 10^4$ cells per well in a 96-well cell culture plate. The following day, cells were infected with SARS-CoV-2 for 30 min at an MOI of 0.1 for Wuhan strain and MOI of 1 for Omicron strain. Cells were washed with PBS and incubated in fresh medium for 24 hours. The medium with or without $1 \times 10^5$ or $2 \times 10^5$ CTLs were added to the cells. Cells were incubated for an additional 24 hours. Supernatants were collected and stored at −80 °C after cell debris were removed by centrifugation at 800 g for 5 min. To measure the amount of viral RNA amplified in Calu-3 cells, the cells were washed three times with PBS and cell-lysis and cDNA synthesis were performed using SuperPrep II Cell Lysis & RT Kit for qPCR (TOYOBO) according to the manufacturer's instructions. To measure the amount of infectious viral particles released from infected Calu-3 cells, 10 μl of the supernatants were incubated with VeroE6/TMPRSS2 (ATCC) cells seeded in a 96-well cell culture plate for 24 h. After washed three times with PBS, cells were lysed and cDNA was synthesized using SuperPrep II Cell Lysis & RT Kit for qPCR (TOYOBO) according to the manufacturer's instructions.

## Antigenic peptide screening

Immune Epitope Database (IEDB) was used for the antigenic peptide screening (https://www.iedb.org).

## Peptide competition assay

Direct association of the $M_{198-206}$ peptide with HLA-A*24:02 was examined using the components of QuickSwitch Quant Tetramer Kit-PE (TB-7302-K1) according to the manufacturer's instructions. Briefly, HLA-ABC Magnetic Capture Beads were mixed with or without QuickSwitch Tetramer or $M_{198-206}$ tetramer. Then, the beads were rinsed and stained with FITC-labeled Exiting Peptide antibody. The beads were rinsed and subjected to flowcytometry analysis using BD LSRFortessa (BD Biosciences). Mean fluorescent intensity (MFI) of FITC channel were calculated using FlowJo (v10.4.2) (TreeStar). The frequency of Exiting Peptide was calculated as follows; no-tetramer control and QuickSwitch Tetramer control were calculated as 0 and 100 percent, individually. Then, generate a linear curve by plotting the MFIs obtained with two controls against percent Exiting Peptide. Finally use the MFI of $M_{198-206}$ tetramer for calculating the percentage of peptide exchange.

## TCR analysis

For the repertoire analysis, total RNA from SARS-CoV-2-$M_{198-206}$ specific cell line was purified and subjected to next-generation sequencing (Repertoire Genesys Inc., Osaka, Japan). cDNAs of TCR alpha and beta chains were linked by T2A sequence and subcloned into pMX-IRES-GFP vector. Ecotropic 293 T cells were used as packaging cells and resulted retroviral supernatant was collected. Viral transduction of the genes to

TG40/CD8 cells were performed as previously described with minor modifications[67,69].

## Single-cell transcriptome analysis

Single-cell libraries were prepared with reagents and instructions from 10x Genomics. cDNA was amplified for 14 cycles, and up to 50 ng of cDNA were used for gene expression libraries. Doublets were removed by using Scrublet[70]. The top 4000 highly variable genes were selected, and used for clustering. Further data analysis was done with BBrowser platform (version 3.3.6, Bio Turing). Cytotoxicity signature and exhaustion signature scores were generated using published lists of genes[37] (Supplementary Table 4) with BBrowser[71]. Dimensionality reduction was done by UMAP (uwot package: https://github.com/jlmelville/uwot.), the number of neighbors is set at 30. Louvain clustering on the PCA results was run by igraph package[72] with a flexible number of nearest neighbors. To detect marker genes, a nonparametric Venice method was used[73]. Venice was also utilized for differential expression analysis between two groups. For trajectory analysis, information about cell embeddings on UMAP were fed to *monocle*3's algorithm to obtain graph's structure[74].

## AIM assay

AIM assay was performed as previously described[20]. PBMCs were cultured for 24 hours in the presence of peptide (10 μg/ml) or DMSO in 96-wells U bottom plates at $1 \times 10^6$ cells/well. CD69+CD137+ cells in CD8+ T-cell population were detected as AIM+ by flow cytometry.

## Statistical analysis

Comparisons were made using the indicated statistical tests using GraphPad software (version 7.02). Unless indicated, Mann-Whitney or Wilcoxon tests were applied for unpaired or paired comparisons, respectively. For library studies, wells greater than mean + 3 SD of the IFNγ levels for wells cultured with aAPCs (without expressing SARS-CoV-2 viral proteins) were calculated for each subject were considered positive. The percentage of positive library wells is presented as: (number of positive wells/total number of wells) × 100.

## Reporting summary

Further information on research design is available in the Nature Portfolio Reporting Summary linked to this article.

# Data availability

Immune Epitope Database was accessible online (https://www.iedb.org/). scRNA-seq data of SARS-CoV-2-M$_{198-206}$-specific CD8+ T cells generated in this study have been deposited in the Gene Expression Omnibus datasets under accession code GSE209676. The remaining data are available within the paper and Source Data file provided with this paper. Source data are provided with this paper.

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

## Acknowledgements

This work was funded by AMED under Grant Number JP20wm0125002, JP20fk0108454, JP20fk0108104, and JP21fk0108534 (J.G., M.Y., Y.K., J.I., and S.Y.), 2021 Hyogo College of Medicine Strategic Grant (H.O.) and partially by Osaka basic medical research promotion foundation (S.I.). We thank all participants for contributing the samples to this study; Dr. Masafumi Sakagami and Dr. Kazumasa Yoshinaga for introducing the collaborators working at Hyogo Medical University and Kyowa-kai Medical corporation to the study; Dr. Wataru Kamitani for sharing anti-SARS-CoV-1-M antibody; Dr. Masayuki Saijo for providing SARS-CoV-2 RNA; Dr. Yoshihiro Kawaoka for providing the SARS-CoV-2 isolate (UT-NCGM02/Human/2020/Tokyo); Dr. Daisuke Motooka, Dr. Daisuke Okuzaki, and Dr. Yu-Chen Liu for single-cell transcriptome analysis and data processing; Dr. Welkin Johnson for critical reading of the manuscript; Ms. Akie Teratani for the excellent technical supports.

## Author contributions

S.I. and H.O. conceived research plans and wrote the manuscript. H.O. performed most parts of experiments including construction of library assay system with the help of A.S. J.G. and M.Y. performed experiments with SARS-CoV-2 and wrote the manuscript. Y.K. and J.I. organized and supervised the experiments with SARS-CoV-2. S.Y. organized and supervised single cell RNA-seq. H.O. and X.L. performed single cell RNA-seq. J.H., Y.T., N.M., H.M. and K.S. organized clinical study; patient enrollment, sample collection and summarizing clinical data at Hyogo Medical University. T.U. provided critical materials for experiments. J.H. and S.D. organized clinical study; patient enrollment and summarizing clinical data at Kyowa-kai Medical corporation. K.M., F.I., Y.F. and T-P.H. performed patient enrollment.

## Competing interests

Hyogo Medical University has filed for patent protection for the identification of specific epitopes. The authors declare no other competing interests.

## Additional information

[1]Department of Microbiology, Hyogo Medical University, Hyogo, Japan. [2]Research Center for Asian Infectious Diseases, The Institute of Medical Science, The University of Tokyo, Tokyo, Japan. [3]Laboratory of Molecular Immunology, Immunology Frontier Research Center, Osaka University, Suita, Japan. [4]Department of Infection Control and Prevention, Hyogo Medical University, Hyogo, Japan. [5]Tokoname City Hospital, Aichi, Japan. [6]Kawanishi City Hospital, Hyogo, Japan. [7]Kyoritsu Hospital, Hyogo, Japan. [8]Kyowa Marina Hospital/Wellhouse Nishinomiya, Hyogo, Japan. [9]Dainikyoritsu Hospital, Hyogo, Japan. [10]Kyoritsu Onsen Hospital, Hyogo, Japan. [11]Joint Research Center for Human Retrovirus Infection, Kumamoto University, Kumamoto, Japan. [12]Department of Emergency and Critical Care Medicine, Hyogo Medical University, Hyogo, Japan. [13]Division of Molecular Virology, Department of Microbiology and Immunology, The Institute of Medical Science, The University of Tokyo, Tokyo, Japan. [14]Research Platform Office, The Institute of Medical Science, The University of Tokyo, Tokyo, Japan. [15]Department of Molecular Immunology, Research Institute for Microbial Diseases, Osaka University, Suita, Japan. [16]Division of Molecular Design, Medical Institute of Bioregulation, Kyushu University, Fukuoka, Japan. [17]Division of Molecular Immunology, Medical Mycology Research Center, Chiba University, Chiba, Japan. ✉e-mail: hi-ogura@hyo-med.ac.jp; sh-ishido@hyo-med.ac.jp

