## [Peer Review File · Nature Communications]

Dysfunctional Sars-CoV-2 M protein-specific cytotoxic T lymphocytes in patients recovering from severe COVID-19Editorial Note: This manuscript has been previously reviewed at another journal that is not operating a transparent peer review scheme. This document only contains reviewer comments and rebuttal letters for versions considered at *Nature Communications*.

REVIEWERS' COMMENTS

Reviewer #1 (Remarks to the Author):

The authors have addressed the remaining comments and updated their claims about the epitope specificity.

Reviewer #2 (Remarks to the Author):

The authors addressed my concerns. Analysis of T cell exhaustion is still done in very few COVID-19 severe patients (3 for single cell analysis) and 6 for tetramer staining and phenotype), but data are consistent.

Reviewer #4 (Remarks to the Author):

The association with this "exhausted" phenotype, as in HIV disease, is likely a consequence of heightened immune activation that is associated with more severe disease. It is unlikely that it is the cause of worse disease in this acute response to what is essentially a neo-antigen. The timing still remains an issue. They found no correlation between phenotype and time after infection which is surprising. There really should be some development of the phenotypes with time. The explanation of the CTL assays remains confusing. Thus, I agree with reviewer 1 that the data fall short of supporting the authors' claim. I also agree with Reviewer #3's concern of novelty and overall advance that this study presents. The fact that they actually could detect responses to many other epitopes further reduces the impact of this paper because it seems they chose to focus on a single epitope rather than explore the T cell response more comprehensively.

Response to reviewer #1

The authors have addressed the remaining comments and updated their claims about the epitope specificity.

We deeply appreciate the comments of reviewer #1 to our responses.

Response to reviewer #2

The authors addressed my concerns. Analysis of T cell exhaustion is still done in very few COVID-19 severe patients (3 for single cell analysis) and 6 for tetramer staining and phenotype), but data are consistent.

We would like to deeply thank reviewer #2 for the valuable concerns and comments regarding the requirement for accurate characterization of exhausted T cells. These comments definitely improved our manuscript. The issue of the sample size pointed by reviewer #2 was included in the discussion section.

Response to reviewer #4

We would like to thank reviewer #4 for giving the valuable comments on our manuscript. We have made several rewritings accordingly to the comments.

The association with this “exhausted” phenotype, as in HIV disease, is likely a consequence of heightened immune activation that is associated with more severe disease. It is unlikely that it is the cause of worse disease in this acute response to what is essentially a neo-antigen.

Thank you for pointing out this important issue. Flow cytometry and scRNA-seq data indicated the “exhausted” phenotype together with less-differentiated phenotype of SARS-CoV-2-M specific CTLs in the convalescents of severe COVID-19 (Fig. 4C), therefore, we postulate that it might be the cause of severe disease. However, as reviewer #4 suggested, it is certainly possible to be a consequence of heightened immune activation that is associated with severe COVID-19. We have now included these points in discussion section.

The timing still remains an issue. They found no correlation between phenotype and time after infection which is surprising. There really should be some development of the phenotypes with time.

We are afraid that reviewer #4 might have misread the manuscript regarding the subject samples. All the patient samples were drawn at the time of convalescent phase but not of acute phase, as described in the methods section headed by “Study subjects and ethics”. Based on that, Fig. 4c and associated Supplementary Fig. 8a showed that the different phenotype of SARS-CoV-2-M₁₉₈₋₂₀₆-specific CD8⁺ T cells between moderate and severe convalescents were independent of time post-onset. Also, Supplementary Fig. 8e showed the frequency of PD-1⁺TIGIT⁺ cells amongst the SARS-CoV-2-M₁₉₈₋₂₀₆-specific CD8⁺ T cells along with “the time post viral clearance”.

The explanation of the CTL assays remains confusing. Thus, I agree with reviewer 1 that the data fall short of supporting the authors' claim.

Original reviewer had raised concern regarding the cytotoxic function of phenotypically exhausted SARS-CoV-2-M specific CD8⁺ T cells from severe convalescents. Therefore, we tried to expand the CD8⁺ T cells from severe convalescents, unfortunately, we could not adequately expand them for the cytotoxicity assay. Hence, we showed that M₁₉₈₋₂₀₆-specific CD8⁺ T cells from moderate convalescents had cytotoxic activity against SARS-CoV-2 infected Calu-3 cells harboring HLA-A*24:02, showing that the cells were bona-fide cytotoxic T lymphocytes. This process was described in the second paragraph of the results section headed by “Identification of an immunoprevalent CTL epitope M₁₉₈₋₂₀₆” .

I also agree with Reviewer #3's concern of novelty and overall advance that this study presents. The fact that they actually could detect responses to many other epitopes further reduces the impact of this paper because it seems they chose to focus on a single epitope rather than explore the T cell response more comprehensively.

We would like to thank reviewer #4 for raising this important issue. We focused on a single epitope M₁₉₈₋₂₀₆ because that is the dominant one in our cohort (Fig. 1c, 1d, 1e, 2b, 2c). In fact, the frequency of the M₁₉₈₋₂₀₆-specific cell was quite high, compared with the other M-related epitopes and the epitopes previously reported as “immunodominant” (Fig.

2b, 2c, Supplementary Fig. 8d). Additionally, they were continuously detected in moderate/severe COVID-19 convalescents from late 2020 through early 2022, when SARS-CoV-2 strains from Wuhan to Omicron spread in our cohort (Fig. 4g) that led us conclude it is an immunoprevalent epitope worth analyzing. However, we recognize the limitation of our scope from the point of view of exploring comprehensive T cell response. The point was included in the discussion section.